# Temperature Dependence of the Thermo-Optic Coefficient of SiO_2_ Glass

**DOI:** 10.3390/s23136023

**Published:** 2023-06-29

**Authors:** Gaspar Rego

**Affiliations:** 1ADiT-LAB, Instituto Politécnico de Viana do Castelo, Rua Escola Industrial e Comercial Nun’Álvares, 4900-347 Viana do Castelo, Portugal; gaspar@estg.ipvc.pt; 2Center for Applied Photonics, INESC TEC, Rua Dr. Roberto Frias, 4200-465 Porto, Portugal

**Keywords:** silica glass, refractive index, material dispersion, thermo-optic coefficient, cryogenic temperatures

## Abstract

This paper presents a thorough analysis on the temperature dependence of the thermo-optic coefficient, d*n*/d*T*, of four bulk annealed pure-silica glass samples (type I—natural quartz: Infrasil 301; type II—quartz crystal powder: Heraeus Homosil; type III—synthetic vitreous silica: Corning 7980 and Suprasil 3001) from room temperature down to 0 K. The three/four term temperature dependent Sellmeier equations and respective coefficients were considered, which results from fitting to the raw data obtained by Leviton et al. The thermo-optic coefficient was extrapolated down to zero Kelvin. We have obtained d*n*/d*T* values ranging from 8.16 × 10^−6^ up to 8.53 × 10^−6^ for the four samples at 293 K and for a wavelength of 1.55 μm. For the Corning 7980 SiO_2_ glass, the thermo-optic coefficient decreases monotonically, from 8.74 × 10^−6^ down to 8.16 × 10^−6^, from the visible range up to the third telecommunication window, being almost constant above 1.3 μm. The Ghosh’s model was revisited, and it was concluded that the thermal expansion coefficient only accounts for about 2% of the thermo-optic coefficient, and we have obtained an expression for the temperature behavior of the silica excitonic bandgap. Wemple’s model was also analyzed where we have also considered the material dispersion in order to determine the coefficients and respective temperature dependences. The limitations of this model were also discussed.

## 1. Introduction

In order to predict the thermal behavior of fiber gratings the knowledge of the thermo-optic coefficients, d*n*/d*T* (also defined as (1/*n*)(d*n*/d*T*)), of the core and cladding materials is required [1,2,3,4]. Vitreous silica is the main component of optical fiber, and it has been an important topic of research for decades. It is well known that the refractive index of glass depends on the wavelength and temperature [5,6], on the presence of intrinsic impurities such as OH^−^ and Cl^−^ [7,8], and on the thermal history, that is, on its fictive temperature [5,9]. Nevertheless, a wide range of values can be found in the published literature, ranging from 7.5 × 10^−6^ to around 11.5 × 10^−6^ [6,8,10,11,12,13,14,15,16,17,18,19]. On the other hand, the core of standard fibers is doped with GeO_2_ and specialty fibers are also co-doped with B_2_O_3_ [20,21]. The refractive index and thermo-optic coefficient of the fiber core can be obtained by using the additive model [22,23], Ghosh’s model [14,24,25] and Wemple’s model for binary and ternary silica-doped glass [26,27,28,29]. Thus, the simulation of the thermal behavior of a fiber Bragg grating (FBG) at room temperature, requires that one knows the fiber parameters, the FBG characteristics and the correct value of the thermo-optic coefficient of the silica cladding, since it impacts the determination of d*n*/d*T* for the core material. Although coatings can be used to change the FBG temperature sensitivity [30], in the present context we consider that the acrylate polymer coating has no effect on the glass temperature sensitivity, as demonstrated in a comparison using FBGs inscribed in bare fibers and through the coating [31]. As far as the optical fiber thermo-optic coefficients are concerned, there is a topic requiring further attention: the glass thermal history. In general, the refractive index is measured by using well annealed bulk samples through the prism angle deviation [15,32,33]. However, the optical fiber fabrication process leaves the core and cladding material in a quenched state, having different fictive temperatures, T_F_ [34,35,36]. It is also known that the refractive index depends on the fictive temperature. In Bruckner’s work [5] it is stated that Type I and Type II silica glass has an anomalous behavior for temperatures below 1500 °C, where the refractive index increases with increasing T_F_. Above 1500 °C it follows the general behavior of doped silica glass, it decreases with the increase in the fictive temperature [37]. Other authors [38,39] claim that they did not find this turning point in the silica glass until 1600 °C, which is in accordance with the behavior of Type III glass at 1700 °C [5]. Comparisons between quenched and annealed samples can be found in the literature. Fleming and Shiever [9] found that for silica glass the refractive index is higher for quenched samples than for the annealed ones, while for SiO_2_-B_2_O_3_ [21] and for SiO_2_-GeO_2_ [40] precisely the opposite was found. Studies on the fabrication of long period fiber gratings (LPFGs) by using the electric arc technique [41,42,43] and CO_2_ laser radiation [44] leads to the conclusion that the cladding refractive index increases after the fabrication procedure, which also agrees with the results obtained for LPFGs inscribed in pre-annealed fibers [45,46]. Therefore, if for FBGs’ simulation the fiber can be considered in a quenched state, the same cannot, in general, be said for simulations of LPFG produced by arc-discharges or CO_2_ laser radiation where the fiber is annealed or partially annealed.

In this paper we have analyzed the temperature dependence of the thermo-optic coefficient of four annealed silica glass samples, from room temperature down to 0 K. We have considered the three-term temperature dependent Sellmeier equation presented by Leviton et al. [47,48,49,50] and we have extrapolated the thermo-optic coefficient down to zero Kelvin. We have obtained the refractive index and d*n*/d*T* at room temperature and for a wavelength of 1.55 μm. We have also revisited the Ghosh [25] and Wemple models [26] for the refractive index and thermo-optic coefficient.

## 2. Determination of the Thermo-Optic Coefficients of SiO_2_ Samples

The experimental data underpinning this work was obtained by Leviton et al. [47,48,49,50] through measurements performed by the method of minimum deviation using the Cryogenic, High-Accuracy Refraction Measuring System (CHARMS) [51,52]. This technique enables the determination of the thermo-optic coefficients with an uncertainty in the order of 2 × 10^−8^ [47]. Therefore, for the Corning 7980 sample the refractive index values were measured for the temperature range from 30 to 300 K and for the wavelength range from 0.4 to 3.6 μm. The experimental data was fitted to a temperature and wavelength dependent Sellmeier expression of the form:(1)$$n^{2}\left(\lambda ,T\right)=1+\sum_{i=1}^{3}\frac{S_{i}\left(T\right)\lambda^{2}}{\lambda^{2}-\lambda_{i}^{2}\left(T\right)},$$
where,
(2)$$S_{i}\left(T\right)=\sum_{j=0}^{m}S_{ij}T^{j},$$
(3)$$\lambda_{i}\left(T\right)=\sum_{j=0}^{m}\lambda_{ij}T^{j},$$
*S_i_*, being the strengths of the resonance features in the material at wavelengths *λ_i_* and ‘m’, is the order of the temperature dependence, in this case is 4 [47].

To determine the thermo-optic coefficient, we performed the derivative in order to temperature of the above equations, resulting:(4)$$2n\frac{dn}{dT}=\lambda^{2}\sum_{i=1}^{3}\frac{\left(\sum_{j=0}^{4}jS_{ij}T^{j-1}\right)\lceil \lambda^{2}-{\left(\sum_{j=0}^{4}\lambda_{ij}T^{j}\right)}^{2}\rceil +2\left(\sum_{j=0}^{4}S_{ij}T^{j}\right)\left(\sum_{j=0}^{4}\lambda_{ij}T^{j}\right)\left(\sum_{j=0}^{4}j\lambda_{ij}T^{j-1}\right)}{\lceil \lambda^{2}-{\left(\sum_{j=0}^{4}\lambda_{ij}T^{j}\right)}^{2}\rceil^{2}},$$

Afterwards, for a wavelength of 1.55 μm we fitted d*n*/d*T* with a 3rd order polynomial, for the given temperature range. In the next step, we have extended the calculation of d*n*/d*T* down to 0 K, since it is known that the thermo-optic coefficient would vanish as it approaches 0 K [5,13,53,54,55]. This conclusion can be drawn from the equations presented by Waxler and Cleek [13] and through the temperature behavior of the thermal expansion coefficient [5]. More recently, Yang et al. [53] showed that d*n*/d*T* follows the temperature behavior of the heat capacity [5]. Moreover, it should be stressed that, at low temperatures, the heat capacity departs from the Debye T^3^ law, it peaks below 20 K before tending to 0 at 0 K [54]. Thus, taking all the information into account, the thermo-optic coefficient can be calculated down to zero Kelvin, by using a 3rd order polynomial, having no independent term. Figure 1 shows, for the wavelength of 1.55 μm, the temperature dependence of the thermo-optic coefficient for Corning 7980 silica glass and the respective refractive index. The latter was obtained through the integration of d*n*/d*T* being the integration constant, determined by knowing that the refractive index at 293 K for 1.55 μm is 1.4444147. It should be noted that the relative error in the determination of d*n*/d*T* at room temperature is less than 0.6% and at temperatures in the order of 30 K is around 3%, since d*n*/d*T* decreases considerably. Nevertheless, this enables us to estimate the values of the refractive index down to zero Kelvin, with the error being in the 6^th^ decimal place. It should be stressed that other approaches were implemented for the sake of comparison such as using, for the refractive index, 6th order polynomials, and through the derivative we calculated the thermo-optic coefficient, or by fitting the refractive index data through a 2nd-order polynomial around the temperature of interest, and through the derivative we found the thermo-optic coefficient. There was a very good convergence of all approaches; that is, the thermo-optic values obtained were essentially the same.

We have proceeded analogously for the other samples: for Heraeus Homosil, data corresponds to the following ranges: 120 to 335 K and 0.34 to 3.16 μm, and the value of ‘m’ in the Sellmeier equation is 3 [50]; for Suprasil 3001, data corresponds to the following ranges: 110 to 310 K and 0.416 to 3.16 μm and ‘m’ is 3 [49]; and for Infrasil 301, data corresponds to the following ranges: 35 to 300 K and 0.5 to 3.6 μm and ‘m’ is 4 [48].

Figure 2 shows the temperature dependence of the thermo-optic coefficient for the four silica samples. Table 1 summarizes the refractive index and thermo-optic coefficient obtained at room temperature (293 K) and 1.55 μm.

The values obtained for the thermo-optic coefficients are similar to the ones published for Heraeus Homosil and Suprasil 3001 samples. Infrasil 301 is slightly lower (1.2%), but Corning 7980 is about 2.5% lower [47,48,49,50]. However, based on the above description, we conclude that our methodology is correct. Moreover, our values adjusted for room temperatures and wavelengths in the visible range compares very well with previously published values for the same type of glass, 8.7–8.8 × 10^−6^ [13,56,57,58,59]. Depending on the physical model used, the thermo-optic coefficient can be associated with different physical parameters. For instance, it can be said that it follows the temperature dependence of the specific heat capacity [53,55] or, as discussed in the next section, that it is related to the temperature shift of the resonance band at ~10.4 eV. In fact, following the dispersion theory, the mean polarizability is associated with the glass resonance bands and changes with both temperature and density. It has been concluded that the density-dependence of the mean polarizability plays an important role in the thermo-optic coefficient of silica glass [60]. Therefore, the differences observed in the properties of the silica samples can be explained by considering that they possess different fictive temperatures which impacts properties such as the refractive index, the thermal expansion coefficient and, namely, the density, which in turn affects the mean polarizability and, therefore, the thermo-optic coefficient.

Figure 3 presents the general behavior of the thermo-optic coefficient as a function of wavelength (which clearly contradicts the results presented by Malitson [6]). It should be stressed that the behavior is temperature dependent (Equation (4)). As it can be seen, d*n*/d*T* changes smoothly with wavelength, where for the scattered data presented in Ref. [47] a non-monotonic decrease can be found. Moreover, for the second and third telecommunication windows, the difference in d*n*/d*T* is less than 0.4%. Note however, that the Heraeus Homosil sample exhibits a slight increase in d*n*/d*T* in the 1.3–1.5 mm region instead [50].

## 3. Ghosh’s Model

Ghosh [14] proposed a model for the thermo-optic coefficient based on the temperature dependence of the excitonic bandgap expressed as follows:(5)$$2n\frac{dn}{dT}=GR+HR^{2},$$
where
(6)$$G=-3\alpha K^{2},$$ α being the thermal expansion coefficient and *K*^2^ = $n_{\infty }^{2}-1\;$ being $n_{\infty }=1.44$ the refractive index at long infrared wavelengths and
(7)$$H=-\frac{1}{E_{g}}\left(\frac{dE_{g}}{dT}\right)K^{2},$$
where *E*g ≈ 10.4 eV is the silica excitonic bandgap and d*E_g_*/d*T* its temperature dependence.
(8)$$R=\frac{\lambda^{2}}{\lambda^{2}-\lambda_{ig}^{2}},$$
accounts for dispersion, where $\lambda_{ig}=0.107\;\mu m$ is the correspondent wavelength of the silica intrinsic bandgap energy at 11.6 eV. Note that for infrared working wavelengths (the second and third telecommunication windows) *R* is essentially unity. Considering the temperature dependence of the thermal expansion coefficient of silica (Figure 4) presented by Okaji et al. [61] and the left side of the equation (Figure 5) obtained in the last section, it is possible to determine *G* and *H* for silica glass. In particular, for Corning 7980 at 293 K, *G* = −1.4098 × 10^−6^ K^−1^ and *H* = 24.8635 × 10^−6^ K^−1^.

As can be observed in Figure 6, at room temperature, *G* is negative and is about 5% of the value of *H* which is positive. In a recent communication by P. Egan [62], from NIST, he summarized the values obtained for the thermal expansion coefficient of the different types of silica glass. At 293 K the values ranged from 4.0 up to 5.2 × 10^−7^ K^−1^, and such a 30% variation accounts for less than 2% change in the thermo-optic coefficient. However, in the last section we observed changes above 5%. Therefore, being the excitonic bandgap essentially intrinsic to silica glass, the changes observed in the thermo-optic coefficients of silica can only be explained by the temperature dependence of the excitonic bandgap that may reflect the thermal history of the glass. By using Equations (5)–(8) and through integration we obtained an expression for the temperature dependence of *E*g = *E*g_0_EXP[f(*T*^6^)], where Eg_0_ was determined in order to give a value of 10.4 at 200 K [63]. Later, the expression was fitted to a 3rd-order polynomial and its derivative to a quadratic polynomial. As observed in Figure 7 the value of *E*g has a slight decrease with the temperature increase being its derivative (Figure 8) at 293 K of about −2.44 × 10^−4^ K^−1^. This value compares well with the one obtained by Ghosh [14]. It should be mentioned that by using Toyozawa’s model [64] with the coefficients given in [63] would result in a thermo-optic coefficient that was 45% larger.

It should be stressed that it was possible to determine the temperature dependence of the excitonic bandgap of silica glass because we knew the refractive index and its temperature dependence. Therefore, bearing in mind that we would like to obtain the cryogenic temperature dependence of the refractive index of germanium doped silica fibers, such as Corning SMF-28, and although we can predict that *H* will be larger than for pure silica, presently we cannot foresee a straightforward way to use Ghosh’s model to reach that goal even if we knew the parameters for pure GeO_2_ at a particular temperature.

## 4. Wemple’s Model

Wemple [26] proposed a model for the refractive index as a function of a minimum number (three) of parameters:(9)$$n^{2}-1=\frac{E_{d}E_{0}}{E_{0}^{2}-E^{2}}-\frac{E_{l}^{2}}{E^{2}},$$
where *E* is the photon energy in electron-volt (*E* = 1.23498/*λ*, *λ* being the wavelength in μm), *E*_0_ is the average electronic energy gap, *E*_d_ is the electronic oscillator strength and *E*_l_ is the lattice oscillator strength. *E*_d_ and *E*_l_ can be estimated through the following expressions:(10)$$E_{d}=fn_{e}Z_{a}\left(N_{A}d^{3}\right),$$
where, for SiO_2_, the coefficient *f* = 5.0, *n*_e_ = 8 equals the number of electrons per anion, *Z*_a_ = 2 is the formal chemical valence of the anion, *N*_A_ is the volume density of anions (Å^−3^) and the dimensionless structure factor *β* = *N*_A_*d*^3^ = 0.184, resulting in *E*_d_ = 14.72 eV.
(11)$$E_{l}=0.86Z_{a}\sqrt{N_{A}/\mu },$$
where *μ* is the reduced mass in atomic mass units. For SiO_2_, replacing *N*_A_^−1^ = 22.7 and *μ* = 7.48, gives *E*_l_ = 0.132 eV. Despite, our working wavelengths being in the second and third telecommunication windows, it is instructive to note that in the visible range of the spectrum *E*»*E*_l_ and, therefore, the second term in the right side of Equation (9) can be neglected, which enables us to write the equation as:(12)$$\frac{1}{n^{2}-1}=\frac{E_{0}}{E_{d}}-\frac{E^{2}}{E_{d}E_{0}},$$

Thus, by plotting *n*^2^ − 1 vs. *E*^2^ it allows us to determine both *E*_d_ and *E*_0_. The plot would result in values of 14.56 and 13.24 for *E*_d_ and *E*_0_., respectively, with the ratio of *E*_d_/*E*_0_ = 1.100. On the other hand, in the far infrared *E*«*E*_0_ and, therefore, Equation (9) can be written as follows:(13)$$n^{2}-1=\frac{E_{d}}{E_{0}}-{\left(\frac{E_{l}}{1.23984}\right)}^{2}\lambda^{2},$$

Thus, by plotting *n*^2^ − 1 vs. *λ*^2^ it is possible to estimate the ratio of *E*_d_/*E*_0_ and *E*_l_. From Figure 9 it is possible to observe that the values obtained depend on the wavelength range used for the fitting. Thus, considering different wavelength ranges above 1.2 μm, one finds that 1.112 < *E*_d_/*E*_0_ < 1.114 and 0.128 < *E*_l_ < 0.134. This way we set limits for the coefficients to be determined. Initially, we have determined the three coefficients by minimizing the difference between Equations (1) and (9), resulting in the following values: *E*_d_ = 15.36, *E*_0_ = 13.89 and *E*_l_ = 0.1206. The fitting was applied from 0.55 μm up to 1.6 μm, with the difference in *n*(@1.55 μm) being in the 5th decimal place. The latter result can be improved such that the difference in *n* follows in the 7th decimal place by fitting in the 1.5–1.6 μm wavelength range (the differences are in the 4th decimal place in the visible range), *E*_d_ = 14.89, *E*_0_ = 13.47 and *E*_l_ = 0.121. Note however, that in both cases the ratio *E*_d_/*E*_0_ and *E*_l_ falls outside the expected limits discussed above. Therefore, in order to improve the accuracy, we have introduced another physical parameter, the material dispersion, which is defined as:(14)$$M=-\frac{\lambda }{c}\frac{d^{2}n}{d\lambda^{2}},$$
where *c* is the speed of light. Thus, *M* was obtained by direct derivative of equation (1) in order to wavelength, to yield:(15)$$M=\left(\frac{-\lambda }{c}\right)\left\{\frac{1}{n}\sum_{1}^{3}\frac{\left(\sum_{j=0}^{4}S_{ij}T^{j}\right){\left(\sum_{j=0}^{4}\lambda_{ij}T^{j}\right)}^{2}\left[3\lambda^{2}+{\left(\sum_{j=0}^{4}\lambda_{ij}T^{j}\right)}^{2}\right]}{{\left[\lambda^{2}-{\left(\sum_{j=0}^{4}\lambda_{ij}T^{j}\right)}^{2}\right]}^{3}}-\frac{1}{n^{3}}\sum_{1}^{3}\frac{\left(\sum_{j=0}^{4}S_{ij}T^{j}\right){\left(\sum_{j=0}^{4}\lambda_{ij}T^{j}\right)}^{2}\lambda }{{\left[\lambda^{2}-{\left(\sum_{j=0}^{4}\lambda_{ij}T^{j}\right)}^{2}\right]}^{2}}\right\},$$
and the result in ps/nm.km (Equation (15) is multiplied by 10^12^) is presented in Figure 10 for a temperature of 293 K.

The zero-dispersion wavelength, λ_0m_, is 1274.34 nm and the material dispersion at 1.55 μm is 22.11 ps/nm.km. On the other hand, through the derivative of Equation (9) it yields [26]:(16)$$M=2169.94\frac{\lambda }{n}E_{l}^{2}+3335.64\left(\frac{E_{l}^{2}}{\lambda E_{d}E_{o}}\right)\left(n-\frac{2}{n}+\frac{1}{n^{3}}\right)-15372\left(\frac{\lambda E_{d}E_{o}}{n}\right)\left(\frac{1}{\lambda^{4}E_{0}^{4}-3.0744\lambda^{2}E_{0}^{2}+2.363}\right)-\;\frac{7882.1}{\lambda }\left(\frac{1}{\lambda^{4}E_{0}^{4}-3.0744\lambda^{2}E_{0}^{2}+2.363}\right)\left(3n-\frac{2}{n}-\frac{1}{n^{3}}\right),$$

It is instructive to note that in the infrared expression can be approximated by:(17)$$nM=2169.94\lambda E_{l}^{2}-15372\left(\frac{E_{d}}{\lambda^{3}E_{0}^{3}}\right),$$
being the relative error in *M* determination of ~0.6%. Furthermore, *E*_l_ can be estimated through the following relation:(18)$$E_{l}=\sqrt{\frac{nM\lambda^{3}}{2169.94\left(\lambda^{4}-\lambda_{0m}^{4}\right)}},$$

Introducing the values above yields 0.132 eV which coincides with the theoretical value obtained using Equation (11) and is in the middle of the expected range for *E*_l_. We fitted *M* in the whole wavelength range, requiring that it should vanish at 1274.34 nm yielding the following set of parameters: *E*_d_ = 22.00; *E*_0_ = 15.10 and *E*_l_ = 0.132. Note that applying the optimum parameters used for *n* optimization would result in a zero-dispersion wavelength of ~0.98 μm. Thus, it can be concluded that it is not possible to fit both equations simultaneously using three fitting parameters in the whole wavelength range. We guess that was the reason for Hammond [27] to propose an equation with four parameters; nevertheless, we will keep only three, but we will also limit the wavelength range from 1.25 up to 1.6 μm such that we can optimize both the zero-dispersion and 1.55 μm wavelengths. Consequently, we obtained the following optimal parameters: *E*_d_ = 14.631, *E*_0_ = 13.188 and *E*_l_ = 0.13184. This way, there is only an error of 5.5 × 10^−4^ in *n*(@1274.34 nm), while at 1.55 μm the difference is in the 6th and 4th decimal places for *n* and *M*, respectively. We have proceeded similarly for the other samples and the results are summarized in Table 2. It should be noted that despite the zero-dispersion wavelength and the material dispersion being at 1.55 μm, which are very similar for all silica samples, the respective temperature dependence can change significantly. Table 3 summarizes the values obtained for the fitting process of all silica samples. It is been stated [65] that the zero-dispersion wavelength shifts linearly with temperature for a wide range of temperatures; however, as can be observed in Figure 11, our results do not comply with this.

Applying Equation (17) at the zero-dispersion wavelength, results in the following [26]:(19)$$\lambda_{0m}=1.6372\sqrt[{4}]{\frac{E_{d}}{E_{0}^{3}E_{l}^{2}}},$$
where the numerical constant was already corrected considering the values obtained in the previous tables.

The temperature dependences of Wemple’s coefficients can be obtained by a derivative of Equations (8), (14) and (17). Thus, one obtains the following equations:(20)$$2n\frac{dn}{dT}=\frac{E_{0}E_{d}^{\prime }}{E_{0}^{2}-E^{2}}-\frac{\left(E_{0}^{2}+E^{2}\right)E_{d}E_{0}^{\prime }}{{\left(E_{0}^{2}-E^{2}\right)}^{2}}-\frac{2E_{l}E_{l}^{\prime }}{E^{2}},$$
(21)$$\frac{dM}{dT}=\frac{4339.88\lambda E_{l}E_{l}^{\prime }}{n}-\frac{2169.94\lambda E_{l}^{2}}{n^{2}}\frac{dn}{dT}+\frac{3335.64}{\lambda }\left(\frac{E_{l}^{2}}{E_{d}E_{0}}\right)\left(\frac{dn}{dT}\right)\left[1+\frac{2}{n^{2}}-\frac{3}{n^{4}}\right]\;+\frac{15372\lambda E_{d}E_{0}}{n^{2}}\left(\frac{dn}{dT}\right)\frac{1}{\left(\lambda^{4}E_{0}^{4}-3.0744\lambda^{2}E_{0}^{2}+2.363\right)}\;-\frac{3335.64}{\lambda }\left(n-\frac{2}{n}+\frac{1}{n^{3}}\right)\left[\frac{E_{l}^{2}E_{d}^{\prime }}{E_{d}^{2}E_{0}}+\frac{E_{l}^{2}E_{0}^{\prime }}{E_{d}E_{0}^{2}}-\frac{2E_{l}E_{l}^{\prime }}{E_{d}E_{0}}\right]\;-\frac{15372\lambda E_{0}E_{d}^{\prime }}{n}\frac{1}{\left(\lambda^{4}E_{0}^{4}-3.0744\lambda^{2}E_{0}^{2}+2.363\right)}\;+\frac{15372\lambda E_{d}E_{0}^{\prime }}{n}\frac{\left(3\lambda^{4}E_{0}^{4}-3.0744\lambda^{2}E_{0}^{2}-2.363\right)}{{\left(\lambda^{4}E_{0}^{4}-3.0744\lambda^{2}E_{0}^{2}+2.363\right)}^{2}}\;-\frac{7882.1}{\lambda }\frac{1}{\left(\lambda^{4}E_{0}^{4}-3.0744\lambda^{2}E_{0}^{2}+2.363\right)}\left(\frac{dn}{dT}\right)\left[3+\frac{2}{n^{2}}+\frac{3}{n^{4}}\right]\;+\frac{7882.1E_{0}^{\prime }}{\lambda }\frac{\left(4\lambda^{4}E_{0}^{3}-6.14884\lambda^{2}E_{0}\right)}{{\left(\lambda^{4}E_{0}^{4}-3.0744\lambda^{2}E_{0}^{2}+2.363\right)}^{2}}\left(3n-\frac{2}{n}-\frac{1}{n^{3}}\right),$$
(22)$$0.55674\lambda_{0m}^{3}\frac{d\lambda_{0m}}{dT}=\frac{E_{d}^{\prime }}{E_{0}^{3}E_{l}^{2}}-3\frac{E_{d}E_{0}^{\prime }}{E_{0}^{4}E_{l}^{2}}-\frac{2E_{d}E_{l}^{\prime }}{E_{0}^{3}E_{l}^{3}},$$

After substitution of the different parameters obtained in the previous tables, the system of three equations and three unknowns yields the solutions summarized in Table 4. As can be observed, the values obtained for *E*’_l_ are very close to the previous estimates. It is also important to note that values can be either positive or negative, with different magnitudes ranging from 10^−4^ to 10^−7^, although, as required, *n* increases with temperature.

Kim and Lines [17] have used two of the three above equations and expressed *E′*_d_ and *E′*_0_ as a function of *E′*_l_. Afterwards, they discussed possible values for these variables, in particular, they demonstrated that the results of Matsuoka et al. [57] do not fit the equations. In our case, only the Infrasil sample would not fit the general behavior. Figure 12 shows our similar analysis for the Corning 7980 sample but using the three equations. As can be observed, both *E′*_d_ and *E′*_0_ are positive values of the same magnitude, ~10^−4^, while *E′*_l_ is negative and approximately one order of magnitude lower. On the other hand, Lines [66] presented a theoretical explanation for the temperature dependence of the variables *E*_d_, *E*_0_ and *E*_l_, and discussed the expected values for *E′*_d_ = −1 × 10^−5^ eV/K, *E′*_0_ = −4.1 × 10^−4^ eV/K and *E′*_l_~−1 × 10^−7^ eV/K. Based on that theory, the thermo-optic coefficient of silica glass would essentially depend on *E′*_0_. However, our results fully contradict this, except for the Suprasil sample where *E′*_0_ is clearly the dominant factor. Moreover, it is expected that, ignoring the temperature dependence of the effective charges, *E′*_l_ = −1.5*αE*_l_ resulting in values around −1 × 10^−7^, but the obtained value for Corning glass is about 100 times larger. Therefore, this analysis following Wemple’s model for the refractive index once again puts in evidence that attempts to standardize the temperature behavior of silica glass may result in misleading conclusions.

## 5. Conclusions

We have analyzed the temperature dependence of the thermo-optic coefficient of four silica glass samples from room temperature down to 0 K. We have obtained dn/dT values ranging from 8.16 × 10^−6^ up to 8.53 × 10^−6^ at 293 K and for a wavelength of 1.55 μm. We observed that, for the Corning 7980 SiO_2_ glass, the thermo-optic coefficient decreases monotonically, from 8.74 × 10^−6^ down to 8.16 × 10^−6^, from 0.55–1.6 μm, being almost constant above 1.3 μm. By using Ghosh’s model, we concluded that the thermal expansion coefficient only accounts for less than 2% of the thermo-optic coefficient value and an expression for the temperature behavior of the silica excitonic bandgap was obtained. The zero-dispersion wavelengths and the material dispersion were determined for all samples. Wemple’s model was also considered, and its limitations discussed. The coefficients and the respective temperature dependencies were determined, and the results were discussed in light of Lines’s theory. For the first time, to the best of our knowledge, a detailed and comprehensive comparison was presented showing that those parameters can change significantly even for pure silica glass samples. The present research will be used to extend the study to Ge-doped silica glass fibers, and results will be published elsewhere.

## Figures and Tables

**Figure 1 sensors-23-06023-f001:**
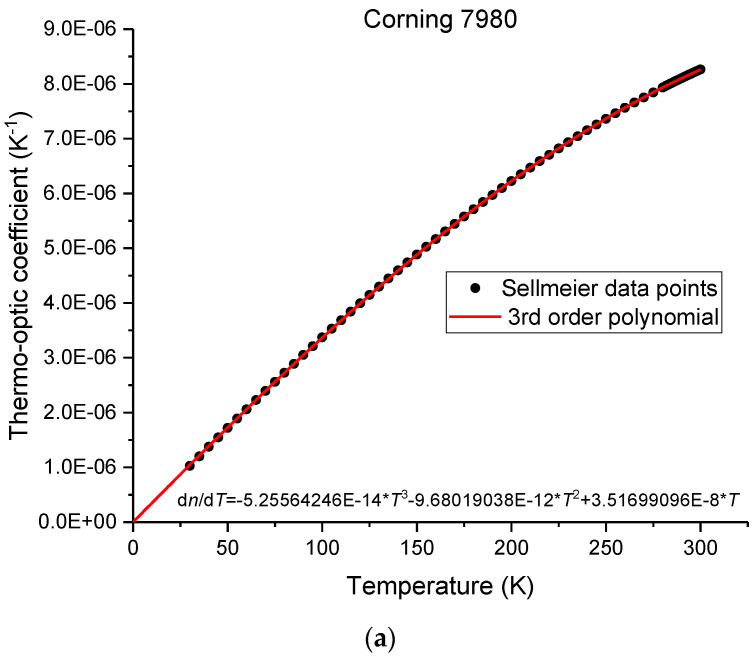
(**a**) Thermo-optic coefficient of the Corning 7980 silica sample and (**b**) refractive index at 1.55 μm, as a function of temperature.

**Figure 2 sensors-23-06023-f002:**
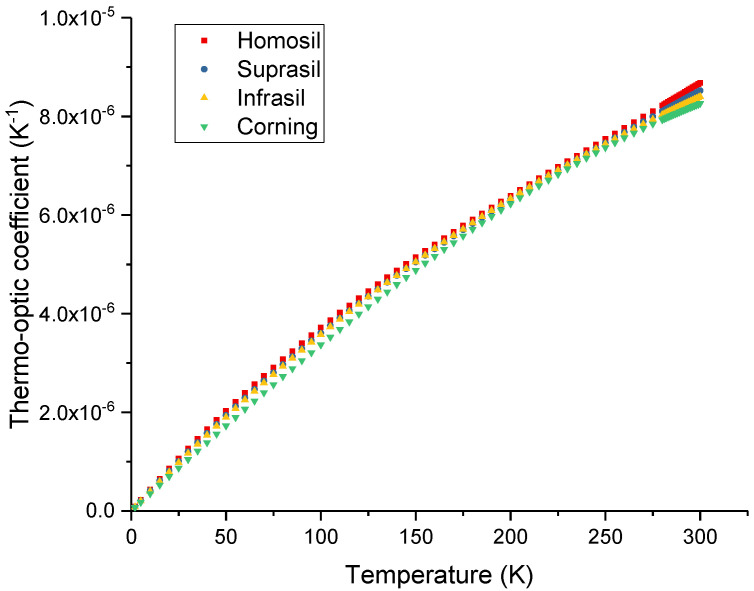
Temperature dependence of the thermo-optic coefficient of the four silica glass samples.

**Figure 3 sensors-23-06023-f003:**
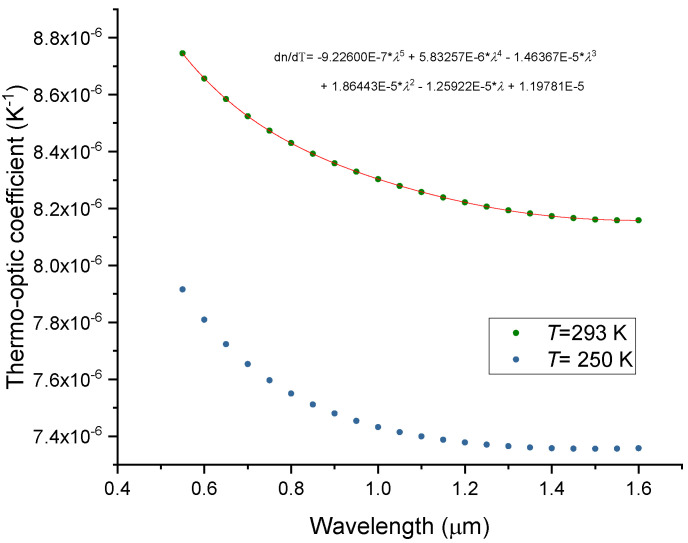
Spectral dependence of the thermo-optic coefficient of the Corning 7980 sample at 250 K and 293 K.

**Figure 4 sensors-23-06023-f004:**
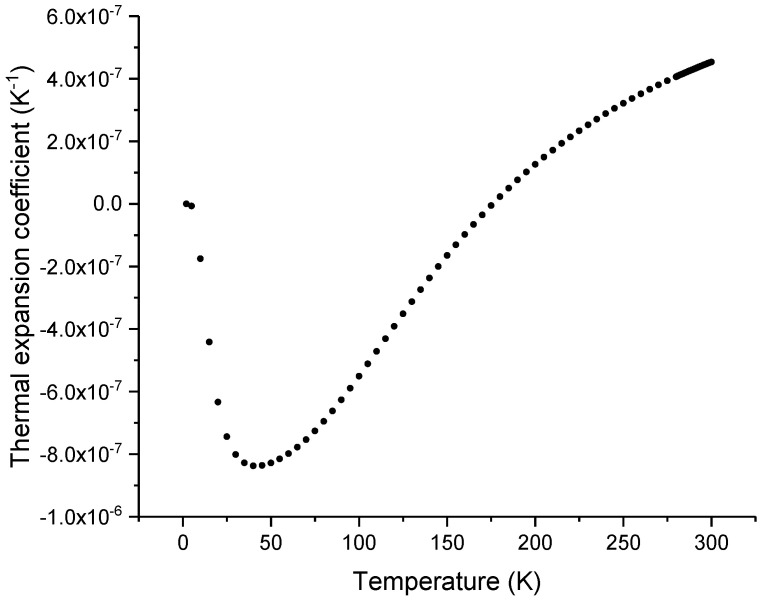
Temperature dependence of the thermal expansion coefficient of silica glass.

**Figure 5 sensors-23-06023-f005:**
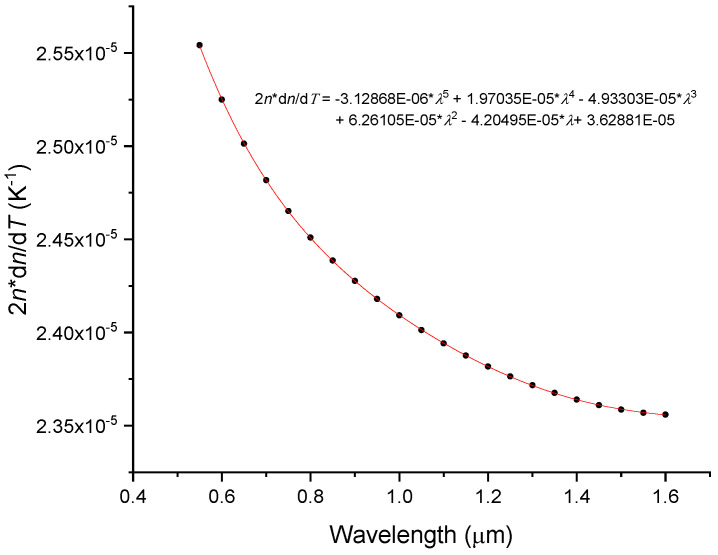
Ghosh’s model at 293 K.

**Figure 6 sensors-23-06023-f006:**
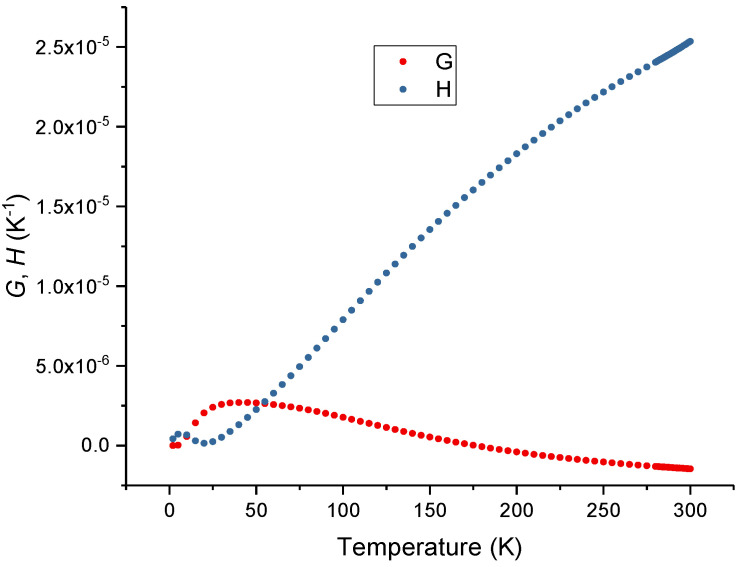
Temperature dependence of *G* and *H* Ghosh’s coefficients.

**Figure 7 sensors-23-06023-f007:**
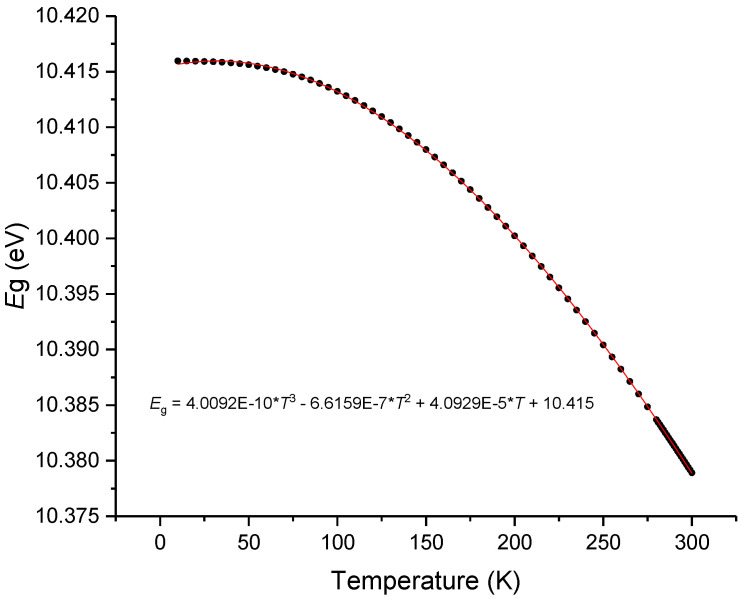
Temperature dependence of the excitonic bandgap of silica glass.

**Figure 8 sensors-23-06023-f008:**
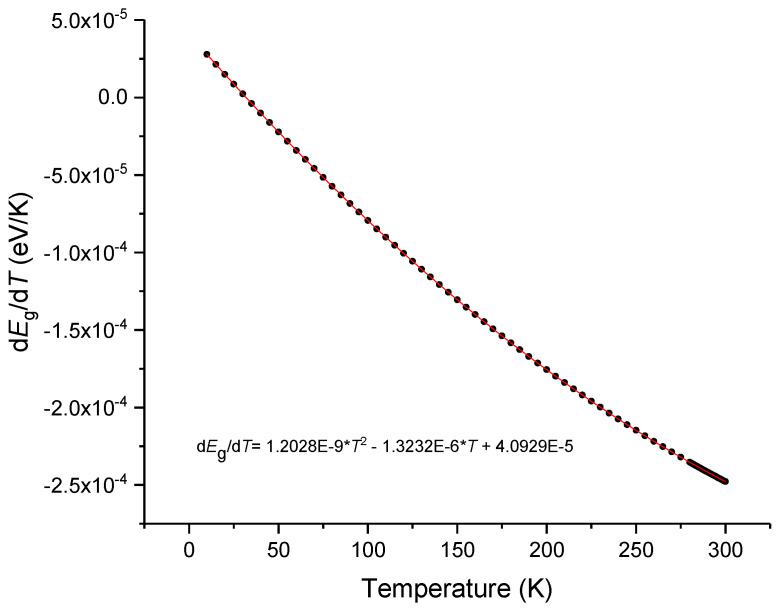
Temperature derivative of the excitonic bandgap of silica glass.

**Figure 9 sensors-23-06023-f009:**
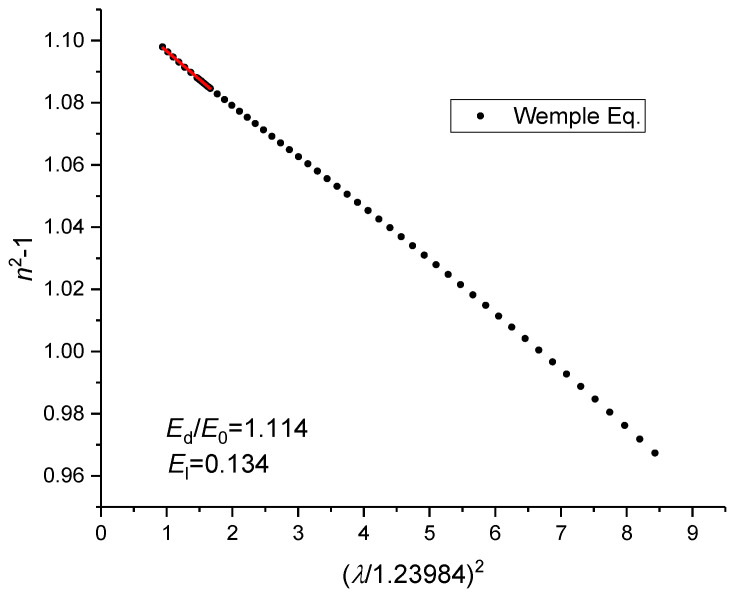
Determination of *E*_d_/*E*_0_ and *E*_l_ in the far infrared range of the spectrum.

**Figure 10 sensors-23-06023-f010:**
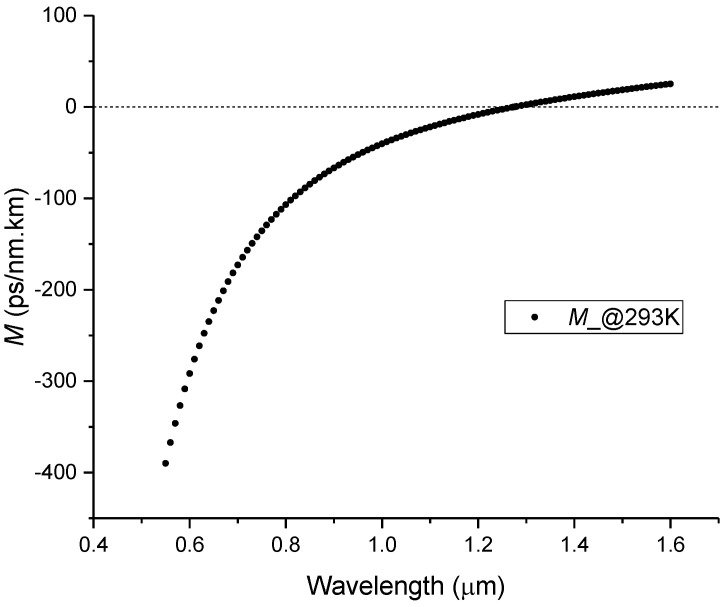
Material dispersion of Corning 7980 sample at 293 K.

**Figure 11 sensors-23-06023-f011:**
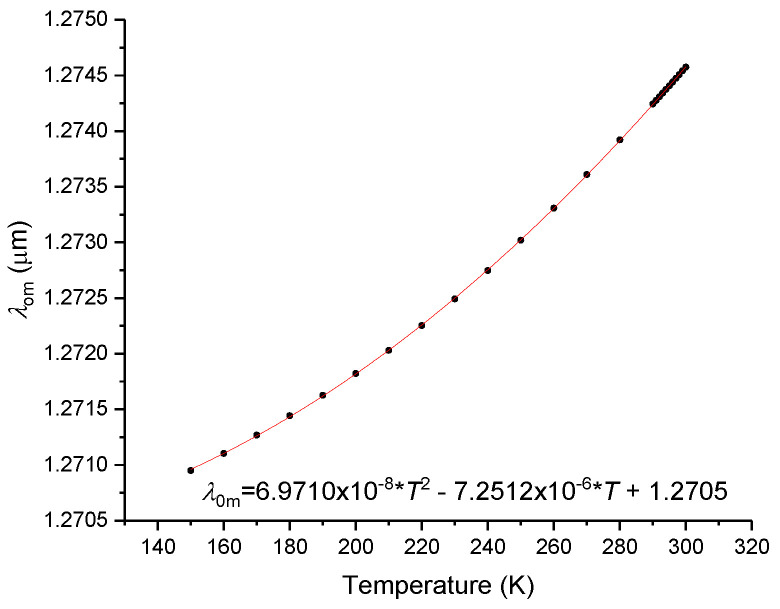
Zero-dispersion wavelength as a function of temperature for Corning 7980 sample.

**Figure 12 sensors-23-06023-f012:**
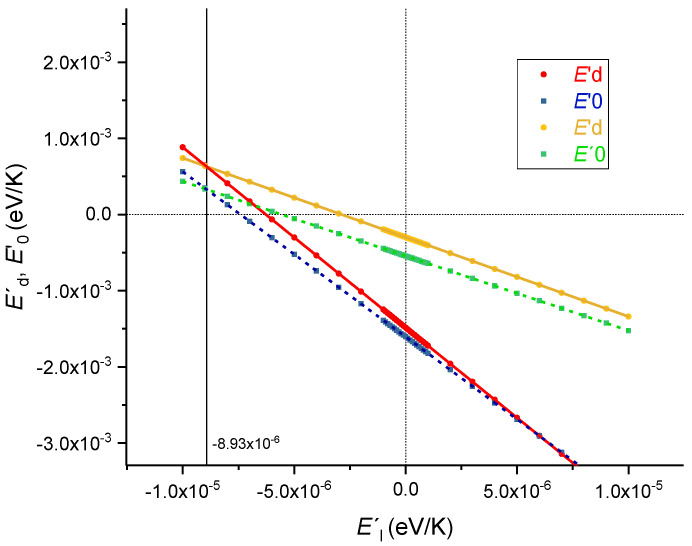
*E′*_d_ and *E′*_0_ as a function of *E′*_l_ for Corning 7980 sample.

**Table 1 sensors-23-06023-t001:** Refractive index and thermo-optic coefficient.

Sample	n	d*n*/d*T* × 10^−6^ K^−1^
Suprasil 3001	1.4446836	8.38
Infrasil 301	1.4445161	8.27
Heraeus Homosil	1.4444717	8.53
Corning 7980	1.4444147	8.16

**Table 2 sensors-23-06023-t002:** Physical parameters at 293 K for *λ*_0m_ and 1.55 μm.

Sample	*λ*_0m_ (nm)	d*λ*_0m_/dT (pm/K)	d*M*_0m_/dT × 10^−3^ (ps/nm.km.K)	*M* (ps/nm.km)	d*M*/dT × 10^−3^ (ps/nm.km.K)
Suprasil	1275.40	22.46	−2.21	21.64	−1.54
Infrasil	1276.17	17.38	−1.71	21.61	−1.06
Homosil	1274.59	23.24	−2.29	21.69	−1.14
Corning	1274.34	32.93	−3.29	22.11	−5.02

**Table 3 sensors-23-06023-t003:** *E*_d_, *E*_0_, *E*_l_ and estimates for *E′*_l_ at 293 K and 1.55 μm by using Equation (18).

Sample	*E*_d_ (eV)	*E*_0_ (eV)	*E*_l_ (eV)	*E′*_l_ (eV/K)
Suprasil	14.755	13.296	0.13056	−4.2 × 10^−7^
Infrasil	14.716	13.266	0.13066	1.6 × 10^−7^
Homosil	14.761	13.309	0.13056	9.7 × 10^−7^
Corning	14.631	13.188	0.13184	−9.1 × 10^−6^

**Table 4 sensors-23-06023-t004:** *E′*_d_, *E′*_0_, *E′*_l_ at 293 K and 1.55 μm.

Sample	*E′*_d_ (eV/K)	*E′*_0_ (eV/K)	*E′*_l_ (eV/K)
Suprasil	−2.1 × 10^−7^	−2.9 × 10^−4^	−4.0 × 10^−7^
Infrasil	5.1 × 10^−5^	−2.4 × 10^−4^	1.8 × 10^−7^
Homosil	−1.6 × 10^−4^	−4.4 × 10^−4^	9.8 × 10^−7^
Corning	6.3 × 10^−4^	3.3 × 10^−4^	−8.9 × 10^−6^

## Data Availability

The data segments can be obtained by contacting the corresponding author.

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
