# Peer review of "Temperature Dependence of the Thermo-Optic Coefficient of SiO2 Glass"

_sensors, 2023, doi:10.3390/s23136023_

Round 1

Reviewer 1 Report

In this manuscript, author demonstrated the experiments for temperature dependence of the thermo-optic coefficient of SiO2 Glass. The study is useful in the application of fiber gratings and other fiber-optic technologies. I enjoyed reading the manuscript because it is clear, well-structured. But I don't think it is suitable for publishing in the journal before adding high-temperature experimental results. Some early literature already has the temperature dependence of quartz glass at about 900k, but the temperature in this article is relatively low. Moreover, the abstract is not suitable for citing references. The word of “we” has been mentioned multiple times in the manuscript, but there is only one author.

Reviewer 2 Report

This manuscript presented a temperature dependence of the thermo-optic coefficient of four bulk annealed pure-silica glass samples from 0 K to room temperature.

I support its publication in this journal due to its probable usage in developing optical fiber sensors in ultralow temperature conditions.

Reviewer 3 Report

The authors present and interesting study to analyze the temperature dependence of the thermo-optic coefficient of four bulk annealed pure-silica glass samples from room temperature down toward 0 K. I find the paper will be of interest for those interested in the field and can be published as is.

Reviewer 4 Report

1. The author should maintain the significance of the coefficient and explain the potential influence on optical fiber at 1550nm.

2. The author should discuss the limitation of the current models (Ghosh model and Wemplel's model).

3. The author should explain why the difference happened for different glass samples.

4. Further emphasize the novelty of this technology instead of just testing and fitting.

5. Evaluate the clarity and relevance of the statement that the present research will be extended to Ge-doped silica glass fibers and the results will be published elsewhere, such as "https://doi.org/10.3390/photonics10040378".

The language is appropriate.

Round 2

Reviewer 1 Report

I agree to accept the current form.